# SOCIALLY-AWARE RECOMMENDER SYSTEMS MITIGATE OPINION CLUSTERIZATION

## ABSTRACT

Recommender systems shape online interactions by matching users with creators' content to maximize engagement. Creators, in turn, adapt their content to align with users' preferences and enhance their popularity. At the same time, users' preferences evolve under the influence of both suggested content from the recommender system and content shared within their social circles. This feedback loop generates a complex interplay between users, creators, and recommender algorithms, which is the key cause of filter bubbles and opinion polarization. We develop a social network-aware recommender system that explicitly accounts for this user-creators feedback interaction and strategically exploits the topology of the user's own social network to promote diversification. Our approach highlights how accounting for and exploiting user's social network in the recommender system design is crucial to mediate filter bubble effects while balancing content diversity with personalization. Provably, opinion clusterization is positively correlated with the influence of recommended content on user opinions. Ultimately, the proposed approach shows the power of socially-aware recommender systems in combating opinion polarization and clusterization phenomena.

## 1 INTRODUCTION

The proliferation of streaming services along with e-commerce platforms has created a need for efficient content Recommender Systems (RS) Linden et al. (2003); Covington et al. (2016); Gomez-Uribe & Hunt (2016). These systems match users with personalized selections drawn from a massive amount of digital content to enhance user experience and ultimately maximize engagement on the platform Li et al. (2024); Raza et al. (2025). On the other hand, by consistently promoting content that aligns with user preferences, RS narrow the diversity of information to which users are exposed, thereby fostering echo chambers and opinion polarization Cinus et al. (2022). Designing RS with high user satisfaction while countering negative global effects such as opinion clusterization has proven challenging despite numerous research efforts Slokom et al. (2025); Su et al. (2013). A key question is how to balance individual user satisfaction with prevention of harmful outcomes at the societal level Lanzetti et al. (2023).

The fundamental principle underlying RS techniques remains consistent: leveraging historical user interaction patterns and profile information to generate personalized recommendations. In this context, collaborative filtering, content-based filtering, and hybrid approaches that integrate both methodologies Li et al. (2024) have emerged as dominant paradigms in the field. However, while personalization enhances user satisfaction at the individual level, it has reinforced negative macroscopic phenomena, such as opinion radicalization Rossi et al. (2022); Lanzetti et al. (2023); Lin et al. (2024). While conventional countermeasures may prove effective in isolated experimental settings, they fail to account for the dynamic response of content creators who strategically adapt their material to target potential audiences Lin et al. (2024); Dean et al. (2024b). Furthermore, most approaches ignore social interactions users have, limiting their capability to operate at a macroscopic level.

Individuals, in fact, do not act in isolation, but are embedded in social contexts where interactions with others influence their preferences and behaviors Proskurnikov & Tempo (2017); Mei et al. (2022). In this regard, a recommendation paradigm arises, known as Social RS, where the social network graph is exploited together with the user-item rating matrix in order to make more accurate

and personalized recommendations Ma et al. (2008); Yang et al. (2014). In contrast to Social RS, our focus is not on neighbours' opinions as a predictor of future preferences. Instead, we aim to dynamically exploit the structure of the social network as a control tool to shape the long-term opinion dynamics in a way that mitigates opinion clusterization.

However, a critical yet underexplored dimension of social RS is how the social network can be leveraged to mitigate harmful content without compromising user satisfaction Hassan (2019). We argue that, opinion polarization being a collective phenomenon, and thus influenced by social interactions, for the RS to mitigate such undesired effects, enhancing content diversity for a single user is not enough. The embedding of the user in a social network is a key component in designing a trustworthy RS Hassan (2019); Chandrasekaran et al. (2024).

This paper addresses a fundamental question: How can RS efficiently leverage the social interactions between users to mitigate global clusterization effects, while simultaneously maintaining high levels of user satisfaction? We show that by explicitly modeling the dynamical interplay of users in the social network, creators' content and the RS, it becomes possible to achieve a better balance between user satisfaction and diversity of opinions.

**Contributions** We propose a novel theoretical framework that models the RS landscape with dynamic interaction between users, content creators, and the RS, where users are embedded in social networks. We leverage the user's network structure to develop an optimization-based RS that mitigates opinion clusterization while maintaining high user satisfaction. Unlike previous approaches, we explicitly model the interaction between users, creators, and the RS, providing a comprehensive solution to the clusterization problem in modern social media platforms. Our main contributions are as follows:

- We propose a framework that captures the dynamic interplay between socially-connected users, strategic content creators, and the RS, and characterize the relationship between content personalization and opinion polarization.
- We show that a RS that greedily optimizes for user satisfaction leads to opinion cluster formation among creators.
- We propose a social-network–aware recommender, $RS(d)$, where the parameter $d$ (number of user hops) controls the trade-off between user satisfaction and the extent of creator clustering, with low d leading to higher satisfaction but more clusters, and high $d$ reducing clusters at the cost of satisfaction.
- We test our algorithm experimentally and showcase that when only accounting for engagement maximization, RS increase opinion clusterization effects over the users population.

**Scope and Limitations** We analyze the relationship between engagement-based recommendation with polarization, under a theoretical framework. The aim is to bring some theoretical insights that can find applications in industrial practice. However, we emphasize that our model is a stylized abstraction. While we believe it captures the core dynamics of real-world recommender systems, actual platforms may exhibit additional behaviors other than confirmation bias due to human irrationality, heterogeneous engagement incentives, or mechanisms beyond confirmation bias, that are not captured by our model.

## 2 RELATED WORK

**Negative impacts of RS.** RS algorithms have been linked to several undesired societal phenomena, including opinion polarization, filter bubbles, and echo chambers. The study Santos et al. (2021) shows that link recommendations between highly similar nodes lead to network topologies that exacerbate opinion polarization. The work in Ziegler et al. (2005) proposes a method to balance diversity and personalization in recommendation lists, enabling exploration of the full spectrum of users' interests and demonstrating improved user satisfaction. Similarly, Cheng et al. (2017); Zhang et al. (2023); Zhang & Hurley (2008) introduce formal optimization frameworks that incorporate diversity objectives into the recommendation process and propose novel metrics to assess diversification quality beyond traditional accuracy measures. These studies collectively show that diversification can be enhanced without severely compromising accuracy, and in some cases even improving it. However, this body of work adopts a static perspective on recommender algorithms, overlooking the dynamic interactions between users and the RS.

**Opinion dynamics.** Opinion dynamics studies how opinions evolve and spread among interacting agents within a social network Proskurnikov & Tempo (2017); Mei et al. (2022); Altafini (2012); Parsegov et al. (2017); Friedkin & Johnsen (1990). In our work, we explicitly account for network influence on user preferences and assume that both users' and creators' preferences evolve dynamically according to an extended version of the Friedkin–Johnsen model Friedkin & Johnsen (1990). This extension incorporates multiple topics Parsegov et al. (2017), where opinions evolve under the joint influence of connected users, recommended content, and each user's own prejudice.

**Link recommendations.** A line of work analyzes the impact of link recommendation over opinions, when opiniond follow a Friedkin-Johnsen dyanmics Wang & Kleinberg (2023); Zhu et al. (2021); Rácz & Rigobon (2023); Kühne et al. (2025); Chitra & Musco (2020). In particular, Wang & Kleinberg (2023) study the impact of addition of links in a social network relates to the level of conflict in the network. Rácz & Rigobon (2023) study how a centralized planner can alter the structure of a social network to reduce polarization. They also analyze the setting where users'internal opinions are adversarially chosen and relate the planner's problem to the maximization of the network Laplacian's spectral gap. The works from Zhu et al. (2021); Kühne et al. (2025); Chitra & Musco (2020) study the design of link recommendations to jointly minimize opinion polarization and disagreement subject to some budget constraints.

**Performative prediction.** Performative predictions support decisions that can influence the outcomes they aim to predict Perdomo et al. (2020). This is the case for RS, whose goal is to predict relevant content for users. The design and evaluation of RS is often approached from a supervised machine learning perspective, treating viewer preferences and the content catalog as static. In practice, however, RS interact with and shape the behavior of both viewers and content creators. This interaction generates a feedback loop between the system and its users Li et al. (2024).

A recent line of work makes this feedback loop explicit by modeling RS–user interactions and studying how users' opinions evolve under the influence of recommended content Dean & Morgenstern (2022); Yao et al. (2024); Rossi et al. (2022); Davidson & Ye (2025); Sprenger et al. (2024); Lin et al. (2024); Dean et al. (2024a); Chandrasekaran et al. (2024).

On the other hand, works such as Ben-Porat & Tennenholtz (2018); Hron et al. (2023); Jagadeesan et al. (2023); Eilat & Rosenfeld (2023); Yao et al. (2023) focus on dynamic adaptation by creators while treating users as static. The position paper from Dean et al. (2024b) proposes a unifying framework that views user–creator–recommender interactions as a dynamical system. In this setting, we draw inspiration from Lin et al. (2024), where both users and creators co-evolve within a feedback loop. Lin et al. (2024) adopt a model-based approach and shows that such dynamics lead to opinion polarization, while standard diversity-promoting strategies are insufficient to mitigate it.

In contrast to Lin et al. (2024), we consider users embedded in a social network, influenced not only by recommended content from creators but also by the content shared from other connected users. We demonstrate that, in this setting, balancing content diversity and personalization counteracts opinion clusterization and polarization.

To better position our paper, the Table in Section G of the Appendix provides a schematic summary of the related work and how we compare with it.

## 3 PROBLEM SETUP

We consider a setup where users interact bidirectionally with content creators, mediated by a RS. Similar to previous works Davidson & Ye (2025); Rossi et al. (2022), opinions are considered as the driving factor behind user preferences. Formally, we consider two sets of agents (modeled as dynamical systems) engaging in $n$ different topics at each time step $t$:

- **Users:** $\mathcal{U}^t = \{u_0^t, u_1^t, ..., u_{N-1}^t\}$, where $u_i^t \in [-1, 1]^n$ represents the opinion of user $i$ at time $t$. The $k$-th entry of $\boldsymbol{u}_i^t$ indicates the opinion of user $i$ on item $k$ at time $t$. We define the global user opinion's vector at time $t$ as $\mathbf{u}^t = \begin{bmatrix} (u_0^t)^\top & \dots & (u_{N-1}^t)^\top \end{bmatrix}^\top$.

- **Content creators:** $\mathcal{C}^t = \{c_0^t, c_1^t, ..., c_{M-1}^t\}$, where $c_j^t \in [-1, 1]^n$ represents the opinion of creator $j$ at time $t$. The $k$-th entry of $c_j^t$ is the opinion of creator $i$ on item $k$ at time $t$. We define the global creator opinion's vector at time $t$ as $\mathbf{c}^t = \begin{bmatrix} (c_0^t)^\top & \dots & (c_{M-1}^t)^\top \end{bmatrix}^\top$.

At each timestep $t$, content creators publish material reflecting their current opinion vectors in $\mathcal{C}^t$. We extend the framework introduced by Lin et al. (2024), and explicitly account for the social network effects in the opinion evolution and overall user behavioral trends. In particular, we assume closed-loop interactions among all agents, creating a dynamic feedback system where: (1) recommended content influences user opinions over time, (2) content creators adapt their content strategies based on audience implicit feedback and (3) users live in a social network where they influence each other over time. While there is a mutual influence between users and creators, the difference between these two classes of agents lies in the nature of their interactions:

- **User-user interactions:** mediated via the social network. This is, two users influence one another if they are connected directly through their social network.
- **Creator-user interactions:** mediated via a RS. Specifically, the RS presents each user with a subset of content creators. In turn, content creators only receive feedback from the users they reach through the RS.

The dynamics resulting from these interactions are modeled as:

$$\mathbf{u}^{t+1} = f(\mathbf{u}^t) + h^t(\mathbf{c}^t), \tag{1a}$$

$$\mathbf{c}^{t+1} = p^t(\mathbf{u}^t) + q(\mathbf{c}^t), \tag{1b}$$

$$c_j^t \sim \mathcal{R}(u_i^t), \qquad \forall i = 1, \ldots, N \quad j = 1, \ldots, M, \tag{1c}$$

where user $i$'s decision of engaging with content from creator $j$ at time $t$, is sampled from a probability distribution $\mathcal{R}$ defined by the recommendation assigned to user's $i$ by the RS selection at time $t$. The functional relationships for $f, h, p, q$ and definitions in equation 1 are given by the multi-topic extended Friedkin-Johnsen model Parsegov et al. (2017) extended to include the influence of the RS in the same fashion as in Sprenger et al. (2024). Detailed descriptions of the functions are discussed in the following sections. For a complete discussion of the model, we refer to Appendix A.

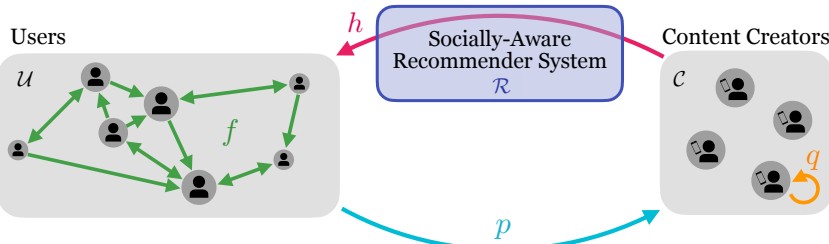

Figure 1: Overview of the dynamic framework described in equation equation 1. Users $\mathcal{U}$ influence each other via dynamics $f(\cdot)$, and are influenced by creators via $h^t(\cdot)$, mediated by $\mathcal{R}$. Content creators have internal opinion dynamics $q(\cdot)$, and are influenced by users via $p^t(\cdot)$.

### 3.1 USER-USER INTERACTION

In our framework, users influence each other by means of interactions through their social network.

**Definition 1** (Social Network). *Let $A \in \mathbb{R}^{N \times N}$ be an adjacency matrix, where $A_{ij} > 0$ indicates that user $j$ influences user $i$ with weight $A_{ij}$. The social network is a directed and weighted graph $\mathcal{G}(\mathcal{U}, \mathcal{E}, A)$, where each node $i \in \mathcal{U}$ corresponds to an individual user, and $\mathcal{E}$ denotes the set of edges representing social connections. An edge $(j, i) \in \mathcal{E}$ exists if and only if user $j$ has a direct influence on user $i$, i.e. $A_{ij} \neq 0$.*

Given Definition 1 and the Friedkin & Johnsen (1990) model for one single topic, i.e. $n = 1$,

$$f(\mathbf{u}^t) = (I_N - \Lambda)A\mathbf{u}^t + \Lambda\mathbf{u}^0, \tag{2}$$

where $I_N$ is the $N$-dimensional identity matrix and $A \in [0, 1]^{N \times N}$ is sub-stochastic and is the adjacency matrix of the social network. The matrix $\Lambda$ is a diagonal matrix whose elements $\lambda_i \in (0, 1]$ capture the resistance to opinion change ("stubbornness") of users, and $u_i^0$ the user's $i$ initial opinion ("prejudice"). Users are assumed a degree of critical thinking, i.e. the diagonal of $A$ is nonzero.

We will adopt the multi-topic version of equation 2 from Parsegov et al. (2017) that is explained in detail in Appendix A.

We note that asymmetric social relationships, common in modern platforms with "following" dynamics, are naturally captured by this representation. More specifically, for each user we can define the set of all users that have an influence on the user via its $d$-hop social network. The parameter $d$ controls the breadth of social context: larger values incorporate more distant connections.

**Definition 2** (*d-hop influencers*). *For a graph $\mathcal{G}(\mathcal{U}, \mathcal{E}, A)$, the d-hop influencers of user $i \in \mathcal{U}$, is $in_i(d) = \{j \in \mathcal{U} \mid dist(j \to i) \leq d\}$, and $i \in in_i(d)$.*

### 3.2 CREATOR-USER INTERACTION

In our framework, creators and users influence each other by means of a RS. In particular, the RS functions as a probability distribution over content creators: for each user $i$, $c_j^t \sim \mathcal{R}(u_i^t)$, where the distribution is dependent on each user's current opinion $u_i^t$. This is, the RS adjusts the probability distribution of content to the user's preferences; details on the specific recommendation strategies studied in this work are provided in the following sections.[1]

**Definition 3.** *Let $\mathcal{F}_1^t, \ldots, \mathcal{F}_M^t$ be the set of disjoint partitions of $\mathcal{U}$ into subsets at time $t$. The user partition $\mathcal{F}_j^t$ contains all users $i$ that consume content from creator $j$ at time $t$, i.e., $i \in \mathcal{F}_j^t$ if $c_j^t$ is sampled from $\mathcal{R}(u_i^t)$ at time $t$.*

This allows for the definition of how content creators and users influence each other.

The influence of content creators towards users follows as

$$h^t(\mathbf{c}^t) = (I_N - \Lambda)B^t\mathbf{c}^t, \tag{3}$$

where $B^t \in [0,1]^{N \times M}$, with $B^t$ such that $[A \; B^t]\vec{1}_{N+M} = \vec{1}_N$ similar to the setting in Sprenger et al. (2024). $B^t$ describes the influence power of the recommendation on user $i$ opinions', where $B_{ij}^t \neq 0$ if $i \in \mathcal{F}_j^t$ at time $t$ and $B_{ij}^t = 0$ otherwise. This is, creator $j$ can only directly influence user's $i$ opinion at time $t$ if they consume their content.

Similar to users, we assume a degree of critical thinking in the creators. This is captured by:

$$q(\mathbf{c}^t) = (I_M - \Gamma)E\mathbf{c}^t + \Gamma\mathbf{c}^0, \tag{4}$$

where $I_M$ is the $M$-dimensional identity matrix and $E, \Gamma \in [0,1]^{M \times M}$ are diagonal matrices (no cross-talk between creators) that govern the temporal consistency of the creator's opinions, determining the influence of their previous stance. $c_j^0$ captures the creator's $j$ initial opinion.

The influence of users towards content creators follows as

$$p^t(\mathbf{u}^t) = (I_M - \Gamma)C^t\mathbf{u}^t, \tag{5}$$

where $C^t \in [0,1]^{M \times N}$, with $C^t$ such that $[E \; C^t]\vec{1}_{M+N} = \vec{1}_M$. $C^t$ describes the influence power of user feedback on creator's $j$ opinions', where $C_{ji}^t \neq 0$ if $i \in \mathcal{F}_j^t$ at time $t$ and $C_{ji} = 0$ otherwise. This is, creator $j$ only receives feedback from the set of users consuming their content.

### 3.3 RECOMMENDER SYSTEM

The RS mediates directly the creator $\to$ user interaction, and indirectly the user $\to$ creator interaction. The goal of a RS is to sort and present content to users to maximize their engagement. According to the confirmation bias theory (Nickerson, 1998), this goal can be achieved by recommending content that perfectly matches the users' existing opinion. However, such a greedy approach was shown to result in polarized opinions and clusterization behavior Del Vicario et al. (2017). Hence, we study the problem of designing a RS that (a) maximizes users' satisfaction while (b) reducing clusterization effects on a global scale. We do so by having the RS explicitly account for the existence of a social network. To that end, we next provide formal definitions of satisfaction and clusterization.

---

[1]A common choice of probability distribution is often the softmax of a utility function that maximizes confirmation bias (Nickerson, 1998). See Anas (1983); Chee et al. (2024); Kalimeris et al. (2021); Hazrati & Ricci (2022) and references therein for additional details.

**Satisfaction.** Satisfaction is quantified by measuring the cumulative distance between a user's opinion and the selected content over the entire time sequence $\{0, ..., T-1\}$.

Motivated by the fact that the engagement of users is driven by confirmation bias Nickerson (1998), the RS will recommend content to maximize user satisfaction, defined as follows.

**Definition 4** (User Satisfaction). *The satisfaction of user $i$ (with opinion $u_i^T \in \mathcal{U}^T$) with creator $j$ (with opinion $c_j^T \in \mathcal{C}^T$) at time $T$ is:*

$$sat(u_i^T, c_j) = \begin{cases} -\frac{1}{T} \sum_{t=0}^{T-1} \|u_i^t - c_j^t\|_2, & if\ c_j^t \sim \mathcal{D}_i^t, \\ 0, & otherwise. \end{cases} \tag{6}$$

Definition 4 formalizes the fact that users' engage more with content that is closely aligned with their own opinion. On a global scale, the RS wants to maximize global satisfaction defined as follows.

**Definition 5** (Global Satisfaction). *Let $\mathcal{U}^t$ be the set of $N$ users at time $t$, the global satisfaction at time $t$ is:*

$$sat(\mathcal{U}^t) = \frac{1}{N} \sum_{i=0}^{N-1} sat(u_i^t). \tag{7}$$

**Clusterization.** Opinion clusterization is quantified by the silhouette coefficient that each user has in its assigned cluster (as computed via $k$-means for the opinion vectors).[2] A high silhouette coefficient indicates that a user's opinion is well-matched to its assigned cluster and poorly matched to neighboring clusters. If users can be clearly assigned to a cluster and thus have high opinion silhouettes, the opinion landscape is clusterized.

**Definition 6** (User Silhouette). *Given an option $u_i^t \in \mathcal{F}_i$, the silhouette coefficient is defined as:*

$$s(u_i^t) = \frac{b(u_i^t) - a(u_i^t)}{\max\{a(u_i^t), b(u_i^t)\}} \in [-1, 1], \tag{8}$$

*where $a(u_i^t) = \frac{1}{|\mathcal{F}_i|-1} \sum_{u_j^t \in \mathcal{F}_i, j \neq i} \|u_i^t - u_j^t\|_2$ is the average intra-cluster distance, and $b(u_i^t) = \min_{l \neq i} \frac{1}{|\mathcal{F}_l|} \sum_{\boldsymbol{u}_k^t \in \mathcal{F}_l} \|u_i^t - u_k^t\|_2$ is the minimum average outer-cluster distance.*

**Definition 7** (Global Clusterization). *The global opinion clusterization of the set $\mathcal{U}^t$ is defined as*

$$cl(\mathcal{U}^t) = \frac{1}{N} \sum_{i=0}^{N-1} s(u_i^t). \tag{9}$$

In what follows, we present a RS design that ensures balance between satisfaction and clusterization by taking into account the social network dynamics.

## 4 RECOMMENDER SYSTEM DESIGN

In this section, we present the design of our *socially-aware RS*. As is standard in personalized feed mechanisms employed by online platforms, we consider a two-stage algorithmic curation strategy that determines which subset of available content reaches each user. At each time step $t$, the RS:

  (i) Computes a reference recommendation $r_i^t \in \mathbb{R}^n$ for each user $i$ as a function of their opinion $u_i^t$, i.e. $r_i^t := r(u_i^t)$.

  (ii) Provides the `top-k` recommendations by presenting the user with the $k$ content items closest to this reference point through $k$-nearest-neighbor search:

$$\mathcal{R}(u_i^t) = \left\{ c_j^t \,\middle|\, \|c_j^t - r_i^t\|_2 \in k\text{-top} \min_j \{\|c_j^t - r_i^t\|_2\}_{j=1}^M \right\}. \tag{10}$$

---

[2]As is standard, the optimal number of clusters is determined by running $k$-means for various values of $k$ and selecting the value that yields the highest average silhouette.

Each user $i$ samples one element (piece of content) from $\mathcal{R}(u_i^t)$ at every time step $t$, i.e., $c_j^t \sim \mathcal{R}(u_i^t)$ according to softmax with temperature parameter $\beta^{-1}$. In what follows, we provide a theoretical result that informs how the choice of $r(u_i^t)$ impacts user satisfaction and clusterization, and provide a design choice that, by mimicking the structure of the social network provably achieves a sweet spot between the two metrics.

## 4.1 RECOMMENDATION STRATEGIES FOR SATISFACTION AND CLUSTERIZATION

By studying the dynamics of the feedback mechanism in 1, it is possible to derive insights on how to design the RS to avoid clusterization while promoting user satisfaction. Consider the dynamics in equation 1 where no RS is intervening and the user partitions are static. In this case, equation 1 becomes:

$$\mathbf{u}^{t+1} = (I_N - \Lambda)A\mathbf{u}^t + \Lambda\mathbf{u}^0 + (I_N - \Lambda)B\mathbf{c}^t, \tag{11a}$$

$$\mathbf{c}^{t+1} = (I_M - \Gamma)E\mathbf{c}^t + \Gamma\mathbf{c}^0 + (I_M - \Gamma)C\mathbf{u}^t, \tag{11b}$$

where $B$ and $C$ are static, since the user partitions are static. This is, users engage with the same content creator over time. In what follows, we explore the influence of the dynamic components (social network $A$, creator's influence $B$, etc.) on the user's opinion evolution.

We also study how different recommendation strategies, influence the overall dynamic behavior and emphasize the key role the RS and the social connections have in this framework. To do that, we consider a recommendation strategy as in equation 10 with $k = 1$. This implies that the RS becomes a deterministic map, i.e. $c_j^t = R(u_i^t)$. Note that this relaxation allows for the bypassing of the stochastic sampling of content given a recommendation – this assumption is solely used for the theoretical results; simulations consider the full stochastic setup.

### 4.1.1 SOCIAL INTERACTIONS MITIGATE THE EFFECT OF RECOMMENDATIONS

**Theorem 1.** *Given the system with dynamics in equation 11, the users' opinions reaches a steady state, i.e., $\exists \mathbf{u}^* \in \mathbb{R}^N$ such that $\lim_{t\to\infty} \mathbf{u}^t = \mathbf{u}^*$. Moreover, the influence of the social network ($A$) towards each user $i \in \{1, \ldots, N\}$, namely $\sum_j A_{i,j}$, and the recommended content, i.e. $B_i$ on $\mathbf{u}^*$ are complementary: increasing one decreases the other.*

The proof can be found in Appendix F.2. Intuitively, Theorem 1 states that when users consistently engage with the same creator over time, they gradually approach the creator's opinion. Creators, in turn, adapt to their fixed audience until an equilibrium is reached. The degree to which this happens depends on the user's and creator's stubbornness (as captured by $\Lambda$ and $\Gamma$, respectively) as well as the strength of the social connections (captured in $A$). The theorem shows that the social network and the recommended content execute two opposing forces over the steady state users' opinions.

We note that the social network promotes dynamics partitions by propagating opinions through user connections. Yet, even connections within the same partition can mitigate the formation of static assignments through indirect network effects. Users within the same partition often maintain paths to users in other clusters, enabling opinion diffusion across partition boundaries.

### 4.1.2 RS THAT MAXIMIZE SATISFACTION INCREASE CLUSTERIZATION

The main goal of a RS is to maximize user's engagement. As such, it is natural to consider RS designs that maximize user's satisfaction. A greedy RS is one that solely maximizes satisfaction by exploiting confirmation bias Nickerson (1998). That is, the RS provides the content creator that most closely aligns with users opinions, i.e.

$$R(u_i^t) = \min_j(||c_j^t - u_i^t||_2). \tag{12}$$

This strategy achieves the maximum satisfaction as per definition of *user satisfaction* in Definition 5.

**Lemma 1.** *Consider the system with dynamics as in equation 1. Let $A$ be a diagonal matrix and creators being stubborn, i.e. $\Gamma = I_M$. Then, a greedy RS as in equation 12 induces a static user partition $\mathcal{F}_1, \ldots, \mathcal{F}_M$ and the users' opinions reach a steady state as per Theorem 1. Moreover, for any user $i \in \mathcal{F}_j$, all opinions approach the recommended creator's content, namely $||u_i^t - c_j^t||_2$ decreases monotonically in time.*

The proof can be found in Appendix F.2. Intuitively, by static assignment of users to creators, users will fall into filter bubbles where they engage with creators who provide users with content that closely aligns with their opinions. This effect is aggravated when the users do not have social interactions to counterbalance the effect of the RS, as per Theorem 1, since static user-creator mappings lead to the formation an equilibrium which fosters clusterization. Moreover, the closer a user's opinion gets to the creators' opinion, the smaller the chances for them to exit the bubble. The critical insight is that global polarization emerges when social network effects are insufficient to counterbalance the RS's homogenizing influence. This tipping point, where clustering effects begin to dominate, depends on the relative strength of social connections versus recommendation influence. We showcase this phenomenon empirically in the experiments section.

### 4.1.3 RS THAT REDUCE CLUSTERIZATION DECREASE SATISFACTION

We have shown that targeting only satisfaction, in the absence of social interactions, promotes opinion clusterization around creators. To counter the static assignment of each user to creators, the RS has to depart from engagement maximization and promote content diversity.

**Corollary 1.** *Let $c_j^t = R(u_i^t)$ in equation 11 $\forall i, j$ result from a non-greedy RS, that is, $R(u_i^t) \neq min_j(||c_j^t - u_i^t||_2)$. Then the user satisfaction in equation 7 is suboptimal.*

*Proof.* The proof follows directly from equation 7. That is $c_k^t = R(u_i^t)$ with $||u_i^t - c_k^t||_2 > min_j ||u_i^t - c_j^t||_2$, for some $j \in 0, ..., M - 1$, will lead to reduced satisfaction of user $i$. $\square$

Together with Lemma 1, this statement promotes the search of a tradeoff between maximizing users satisfaction while countering opinion clusterization.

### 4.2 A SOCIALLY-AWARE RECOMMENDER SYSTEM MIMICS THE SOCIAL NETWORK

Given that social connections naturally mitigate clustering, we propose a recommendation strategy that explicitly leverages network structure. Rather than optimizing solely for individual preferences, our approach expands on the greedy RS in a socially-aware manner. This is, the RS incorporates the opinions of users within a social neighborhood besides the individual opinion of the user. To do so, we will leverage the concept of *d-hop influencers* from Definition 2. In particular, we design a RS that mimics the social influence by leveraging the mean opinion of the $d$-hop influencers of each user, and using this as the RS recommendation reference.

**Definition 8** (*d*-hop socially-aware RS). *Let $in_i(d)$ be the d-hop influencers of user $i$. The d-hop socially aware RS produces a recommendation reference that is given as:*

$$R(u_i^t) = \frac{1}{|in_i(d)|} \sum_{j \in in_i(d)} u_j^t. \tag{13}$$

The design parameter $d$ controls the trade-off between personalization and diversification: $d = 0$ recovers the greedy strategy in Lemma 1, while larger values incorporate broader social influence. Since this approach explicitly mimics social influence mechanisms, we have the following:

**Lemma 2.** *Let $R(u_i)$ be a d-hop socially-aware RS with parameter $d$. Then, global clusterization as per Definition 7 decreases as $d$ increases.*

The proof can be found in Appendix F.3. Intuitively, by the dynamics in equation 11, if the RS provides the $d$-hop neighborhood average opinion then users' opinion are pulled towards the average opinion of their neighborhood, including neighbors from other clusters. This mechanisms will reduce the separation between cluster. Increasing $d$ enhances content diversity at the cost of reduced user satisfaction, as recommendations deviate further from individual preferences. In the next section, we empirically analyze this trade-off and identify parameter values that minimize global clustering while maintaining user satisfaction.

## 5 EXPERIMENTS

We evaluate the performance of our RS using the satisfaction and clusterization metrics as in equation 5 and equation 7. In a first experiment we study the closed loop interaction of our RS with a

synthetic social network. We then demonstrate our findings on the Facebook-ego dataset McAuley & Leskovec (2012), which comprises the social network of $4039$ users. Throughout this section, the temperature parameter governing the users choice is chosen as $\beta^{-1} = 0.5$. We present our main findings below.

### 5.1 EXPERIMENTAL SETUP FOR SYNTHETIC DATASET

We consider a network of $N = 600$ users and $M = 50$ content creators, with an average user in-degree of $11$. All interaction parameters governing the user-creator and creator-user dynamics in equation 1 are detailed in Appendix B.1. For clarity of representation, we set the opinion dimension to $n = 2$, with all user and creator opinions initialized uniformly at random within $[-1, 1]^n$. The social network topology is randomly generated, with the probability of an edge of being present between users decreasing with their opinion distance. As a result, users with closer opinions are more likely to be connected. This setup aligns with the homophily principle in social networks, which leads to more contact between similar users McPherson et al. (2001). The specific connection probability function, is detailed in Appendix B.2.

### 5.2 EXPERIMENTAL RESULTS ON SYNTHETIC DATASET

The RS follows a `top`-$k$ recommendation strategy with $k = 5$. The impact of different $k$ and varying interaction parameters in equation 1 on all performance metrics for the following settings is analyzed in Appendix C.

#### 5.2.1 CLUSTERIZATION AND SATISFACTION WITH DIFFERENT RS STRATEGIES

We investigate three distinct RS strategies: (i) RS only optimizes for user engagement (greedy RS with $d = 0$), (ii) RS only accounts for opinion diversity (*socially aware RS* with $d = 6$, high diversification), and (iii) hybrid RS that balances satisfaction and diversity (socially aware RS with $d = 3$, intermediate localization). Strategy (i) only accounts for personalized recommendations, strategy (ii) emphasizes social interactions by considering broad network effects, while strategy (iii) seeks trades-off between user satisfaction and content diversity.

Figure 2 shows the evolution of user and creator opinion landscapes at different time steps under the three proposed RS strategies. User opinion clusters, obtained via k-means clustering, are visualized as ellipsoids with axes corresponding to twice the standard deviation along the principal components. The ellipsoid transparency decreases with global clusterization. Clusters with global clusterization values below 0.5 are omitted, as they lack structural definition. The last column in fig. 2 shows the negative global clusterization and user satisfaction over time. Ideally, one would like both curves to increase over time. We notice how the greedy RS (i) leads to the formation of distinct user clusters around creators, with this polarization effect intensifying over longer time horizons. The socially-aware RS (ii) reduces clusterization but at the cost of significantly lower user satisfaction. The hybrid RS (iii) maintains high satisfaction while effectively mitigating clusterization effects, achieving a balance between the two objectives.

Figure 3 displays negative global opinion clusterization and user satisfaction as functions of the localization parameter $d$ at $t = 50$ and $t = 500$, along with their variance[3]. As $d$ increases, both clusterization and satisfaction decline. A sweet spot occurs at $d = 3$, where clusterization drops sharply while satisfaction remains high.

### 5.3 EXPERIMENTAL SETUP FOR REAL DATASET

The ego-Facebook dataset comprises the social network of $4039$ users. The resulting social graph $\mathcal{G}(\mathcal{U}, \mathcal{E}, W)$ has an average degree of $45$ and is bidirectional, i.e. $(i, j) \in \mathcal{E}$ if and only if $(j, i) \in \mathcal{E}$. We consider $M = 120$ content creators with random initial opinions and set the opinion dimension to $n = 3$. All interaction parameters governing the user-creator and creator-user dynamics in equation 1 are detailed in Appendix D.1. To generate initial user opinions that reflect the homophilic structure inherent in social networks, we employ spectral clustering Luxburg (2004) to identify net-

---

[3]Variance is computed over users as in equation 7 and equation 9.

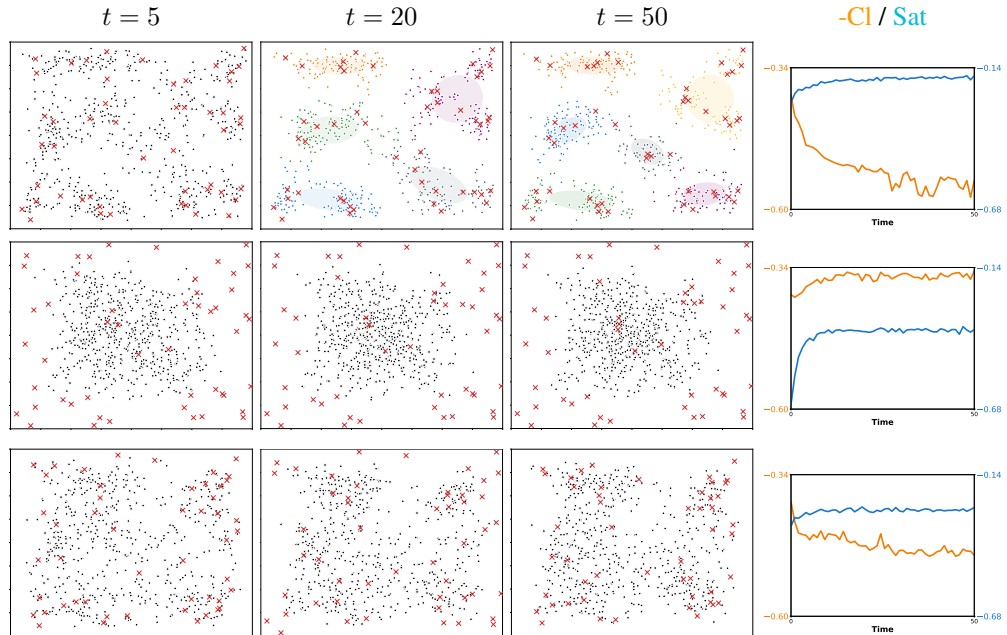

Figure 2: Snapshots of the opinion environment simulated with a localized region (i) $d = 0$, first row, (ii) $d = 6$, second row (iii) $d = 3$, third row. $\times$ denotes the creator, $\bullet$ the users respectively.

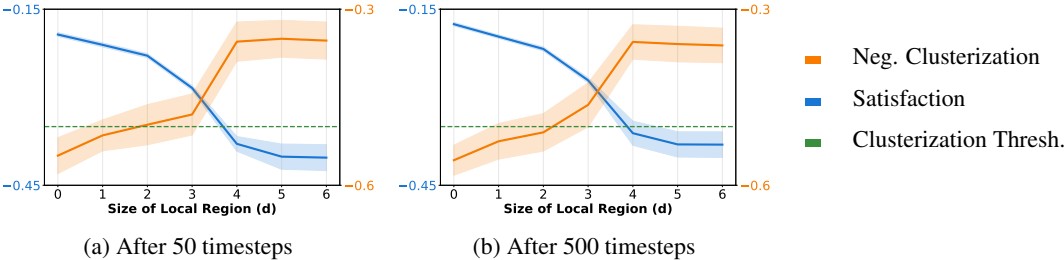

(a) After 50 timesteps        (b) After 500 timesteps

Figure 3: Global clusterization and global user satisfaction plotted as $d$ varies after (a) 50 and (b) 500 timesteps. Clusterization thresh is set to $-0.5$, for which clusters are no longer distinguishable.

work communities, then assign similar opinions to users within the same community. The details can be found in Appendix D.2.

## 5.4 EXPERIMENTAL RESULTS ON REAL DATASET

The RS operates under a $\texttt{top}-k$ recommendation strategy with $k = 5$. Figure 4 displays the user opinion landscapes after $t = 20$ time-steps for (i) a greedy RS with $d = 0$ and (ii) a hybrid RS with $d = 3$. For illustration purposes, we omitted the creators opinions in the 3-dimensional figure on the left side and only display every fourth opinion for the the users and the creators opinions respectively in the figures showing the projections on the $xy$-, $xz$- and $yz$-planes.

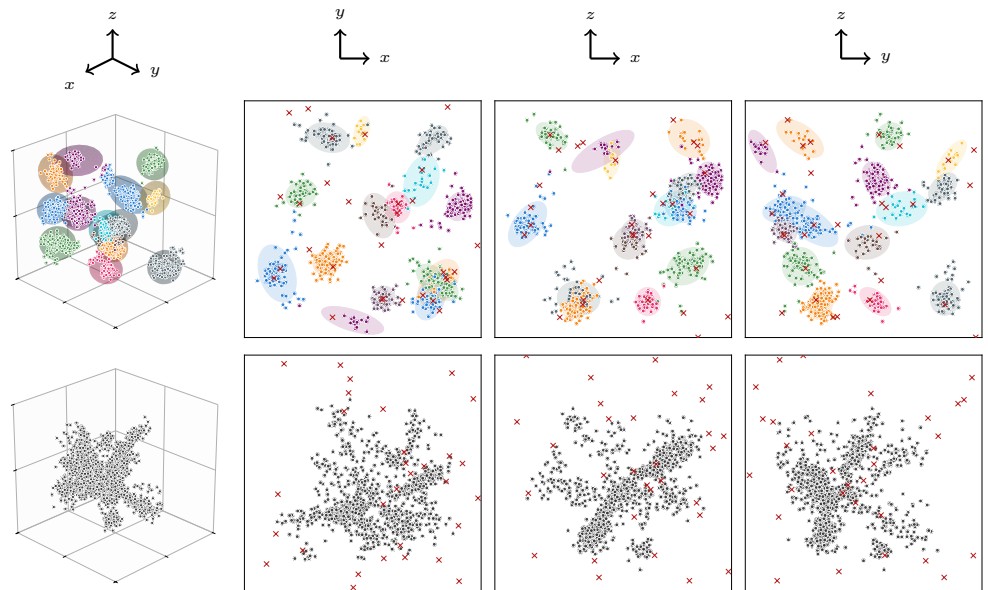

Figure 4: Snapshots after $t = 20$ timesteps of the opinion environment simulated with a localized region (i) $d = 0$, first row, (ii) $d = 3$, second row. $\times$ denotes the creator, $\bullet$ the users respectively.

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

## A  MULTI-TOPIC FRIEDKIN-JOHNSEN MODEL

The multi-topic FJ model Parsegov et al. (2017) extends the classical scalar model in equation 2 to the case where each user holds opinions on multiple topics simultaneously. Let $u_i^t \in \mathbb{R}^n$ denote the opinion vector of user $i$ at time $t$, where each entry corresponds to a distinct topic. Stacking all $N$ users' opinions into a single vector $\mathbf{u} \in \mathbb{R}^{Nn}$, the opinion update rule can be written as

$$f(\mathbf{u}^t) = ((I_N - \Lambda)\hat{A} \otimes C)\mathbf{u}^t + (\Lambda \otimes I_n)\mathbf{u}^0, \tag{14}$$

where $\hat{A} \in \mathbb{R}^{N \times N}$ is the row-stochastic influence matrix describing interpersonal influence in the network, $\Lambda \in \mathbb{R}^{N \times N}$ is the diagonal susceptibility matrix capturing how attached each influence is to their own prejudice $u^0 \in \mathbb{R}^{Nn}$ versus the social influence, and $\otimes$ denotes the Kronecker product. The matrix $C$ is a correlation matrix among different topics. We consider the case of uncorrelated topics, and hence set $C = I_n$. In this special case, the model reduces to

$$f(\mathbf{u}^t) = ((I_N - \Lambda)\hat{A} \otimes I_n)\mathbf{u}^t + (\Lambda \otimes I_n)\mathbf{u}^0. \tag{15}$$

## B    ENVIRONMENT VARIABLES FOR SYNTHETIC DATASET

### B.1    PARAMETERS FOR USER-CREATOR DYNAMICS

The parameters governing the dynamics in equation 1 are sampled independently from uniform distributions with bounds given in Table 1.

Table 1: Simulation Parameters for Uniform Distribution Sampling

| Parameter | Lower Bound | Upper Bound |
|---|---|---|
| *User Parameters* | | |
| User Stubbornness $\Lambda_i$ | 0.0 | 0.5 |
| User Self-Influence $A_{ii}$ | 0.5 | 0.8 |
| Recommender Influence $B_{ij}$ | 0.2 | 0.8 |
| Neighbor Influence $A_{ij}$ | 0.025 | 0.05 |
| *Creator Parameters* | | |
| Creator Stubbornness $\Gamma_j$ | 0.0 | 0.5 |
| Creator Self-Influence $E_j$ | 0.5 | 0.8 |
| User-Creator Influence $C_j$ | 0.2 | 0.8 |

The user-creator influence is evenly distributed among the audience set of creator $j$; specifically, for creator $j$ with audience set $\mathcal{F}_j$, each user $i \in \mathcal{F}_j$ exerts influence $C_{ji} = C_j/|\mathcal{F}_j|$. The overall social influence on user $i$ is determined by summing the influences from all neighbors. Each user is influenced by exactly one creator. Thus, referring to the FJ model in appendix A, we obtain the stochastic constraints: $A_{ii} + B_{ij} + \sum_{j=0}^{N-1} A_{ij} = 1$ for users and $C_j + E_j = 1$ for creators.

### B.2    USER-USER INTERACTION PROBABILITY

For each user $j$, we assume the connection to any other user with a probability that is given as:

$$\text{Prob(user } j \text{ influences user } i) = \exp(-\delta||u_i^0 - u_j^0||_2^2)$$

where $u_i^0, u_j^0 \in [-1, 1]^n$ are the opinion vectors of users $i$ and $j$ at time 0, and $\delta > 0$ is a parameter controlling the connectivity of the network. Different parameters of $\delta$ lead to different number of connections. Different choices of parameter $\delta$ and the resulting average node degrees are displayed in Table 2. We choose $\delta = 9$, to recover 11 neighbors.

Table 2: Network connectivity for different parameters $\delta$ with $N = 600$ users initialized randomly

| Parameter $\delta$ | Average Connections |
|---|---|
| 6 | 21 |
| 7 | 17 |
| 8 | 14 |
| 9 | 11 |

## C    RESULTS FOR VARIATIONAL ENVIRONMENT ON SYNTHETIC DATA

### C.1    VARYING THE SIMULATION PARAMETERS FOR THE DYNAMICS

We increase the number of social interactions by setting $\delta = 6$, yielding an average of 21 connections per user. The expanded social network exposes users to a broader spectrum of opinions through peer interactions. Figure 5a and fig. 5b present the global clusterization and satisfaction metrics after 50 and 500 timesteps, respectively. The results demonstrate that increased social connectivity mitigates clusterization, even under the greedy recommender system ($d = 0$).

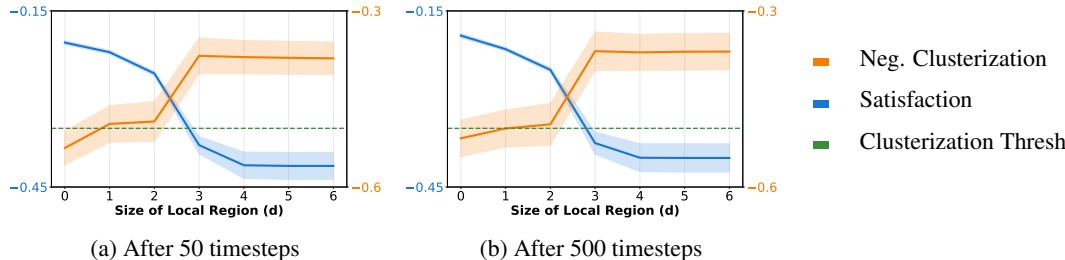

(a) After 50 timesteps        (b) After 500 timesteps

Figure 5: Global clusterization and global user satisfaction plotted as $d$ varies after (a) 50 and (b) 500 timesteps with more social interactions as opposed to fig. 3. Clusterization thresh is set to $-0.5$, for which clusters are no longer distinguishable.

## C.2 VARYING $k$ FOR TOP-$k$

The opinion dynamics under socially-aware recommender systems are examined for localization parameters $d \in \{0, 3\}$ and recommendation set sizes $k \in \{1, 2, 3, 4\}$. Figures 6–9 display the corresponding opinion landscapes. For $k = 1$, the dynamics becomes stationary after $t = 20$ (fig. 6). Cluster visualization is omitted given the low global clusterization score. The number of clusters increases as $k$ decreases, reflecting reduced creator-user interaction diversity. The recommender system with $d = 3$ produces fewer clusters than $d = 0$ across all values of $k$.

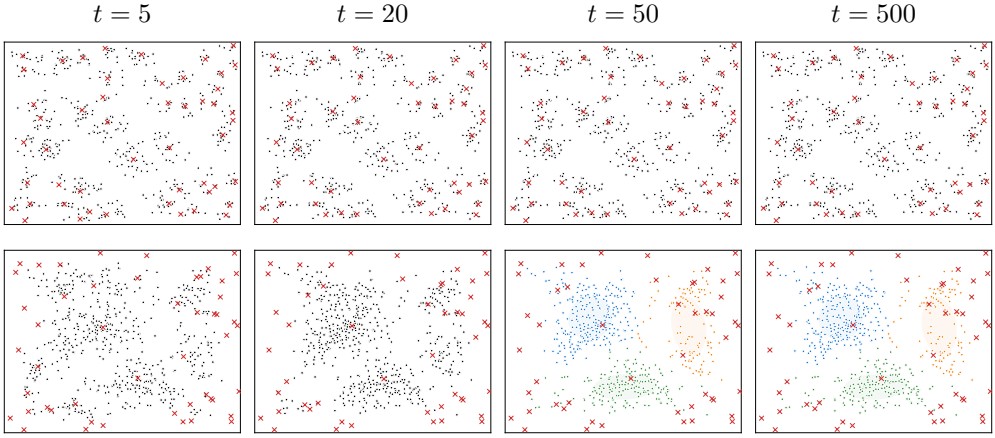

Figure 6: Snapshots of the opinion environment with $k = 1$ simulated with a localized region $d = 0$, first row, $d = 3$, second row. $\times$ denotes the creator, $\bullet$ the users respectively.

## D ENVIRONMENT VARIABLES FOR REAL DATASET

### D.1 PARAMETERS FOR USER-CREATOR DYNAMICS

The parameters governing the dynamics in equation 1 are sampled independently from uniform distributions with bounds given in Table 3.

The user-creator influence is evenly distributed among the audience set of creator $j$; specifically, for creator $j$ with audience set $\mathcal{F}_j$, each user $i \in \mathcal{F}_j$ exerts influence $C_{ji} = C_j/|\mathcal{F}_j|$. The overall social influence on user $i$ is determined by summing the influences from all neighbors. Each user is influenced by exactly one creator. Thus, referring to the FJ model in appendix A, we obtain the stochastic constraints: $A_{ii} + B_{ij} + \sum_{j=0}^{N-1} A_{ij} = 1$ for users and $C_j + E_j = 1$ for creators.

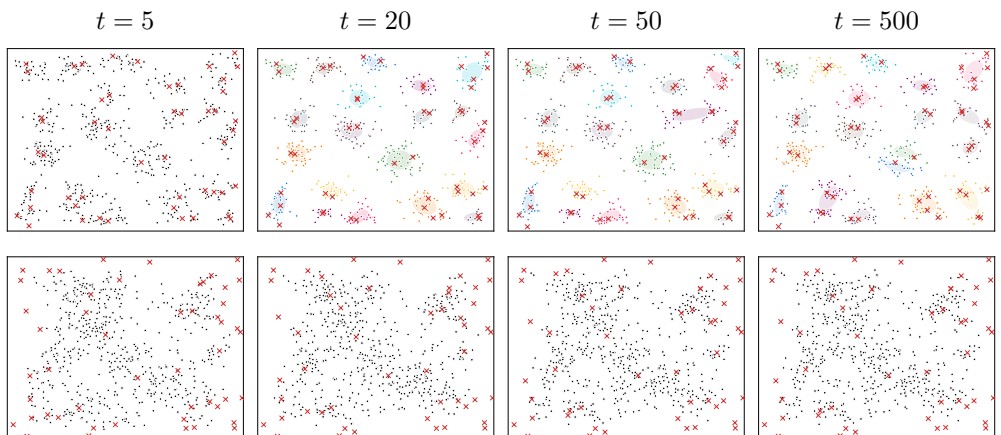

Figure 7: Snapshots of the opinion environment with $k = 2$ simulated with a localized region $d = 0$, first row, $d = 3$, second row. $\times$ denotes the creator, $\bullet$ the users respectively.

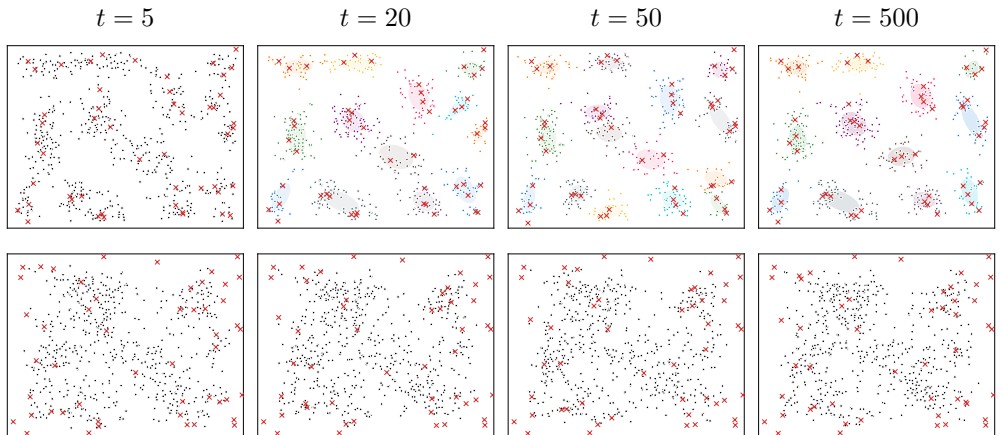

Figure 8: Snapshots of the opinion environment with $k = 3$ simulated with a localized region $d = 0$, first row, $d = 3$, second row. $\times$ denotes the creator, $\bullet$ the users respectively.

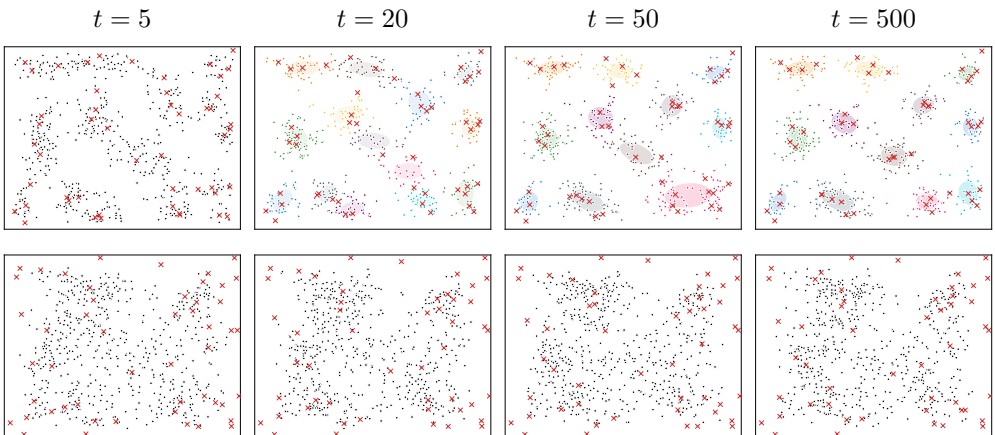

Figure 9: Snapshots of the opinion environment with $k = 4$ simulated with a localized region $d = 0$, first row, $d = 3$, second row. $\times$ denotes the creator, $\bullet$ the users respectively.

Table 3: Simulation Parameters for Uniform Distribution Sampling

| Parameter | Lower Bound | Upper Bound |
|---|---|---|
| *User Parameters* | | |
| User Stubbornness $\Lambda_i$ | 0.0 | 0.5 |
| User Self-Influence $A_{ii}$ | 0.5 | 0.8 |
| Recommender Influence $B_{ij}$ | 0.2 | 0.8 |
| Neighbor Influence $\sum_{j=0}^{N-1} A_{ij}$ | 0.25 | 0.5 |
| *Creator Parameters* | | |
| Creator Stubbornness $\Gamma_j$ | 0.0 | 0.5 |
| Creator Self-Influence $E_j$ | 0.5 | 0.8 |
| User-Creator Influence $C_j$ | 0.2 | 0.8 |

## D.2 EGO-FACEBOOK DATASET

The network comprises 4039 anonymous users and their social connections. The resulting graph has an average degree of 45 with a number of neighbors reaching from 2 to 1046. We identify 34 community centers $\{C_1, C_2, ..., C_{34}\}$ that are randomly dispersed in $[0,1]^3$ and apply spectral clustering to assign each user to one of the specified communities. After assignment, any user $i$, assigned to community center $j$, is initialized with $u_i^0 = C_j + \epsilon_i$, with $\epsilon_i \sim \mathcal{N}(0, 0.15)$.

## E   COMPUTE RESOURCES

All simulations and experiments were conducted on a MacBook Air equipped with an Apple M2 chip and 8 GB of unified memory, running macOS 15.6.1.

## F   THEOREM AND LEMMAS PROOFS

### F.1   PROOF OF THEOREM 1

*Proof.* The proof follows by the extended Friedkin-Johnsen dynamics with

$$\begin{bmatrix} u^{t+1} \\ c^{t+1} \end{bmatrix} = \begin{bmatrix} I - \Lambda & 0 \\ 0 & I - \Gamma \end{bmatrix} \underbrace{\begin{bmatrix} A & B \\ C & E \end{bmatrix}}_{\Pi} \begin{bmatrix} u^t \\ c^t \end{bmatrix} + \begin{bmatrix} \Lambda & 0 \\ 0 & \Gamma \end{bmatrix} \begin{bmatrix} u^0 \\ c^0 \end{bmatrix} \tag{16}$$

with $\Pi \in \mathbb{R}^{(N+M)\times(N+M)}$ row stochastic. From (Proskurnikov & Tempo, 2017, Theorem 21) we get that in the limit for $t \to \infty$ we have

$$\begin{bmatrix} u^\infty \\ c^\infty \end{bmatrix} = \left( \underbrace{\begin{bmatrix} I & 0 \\ 0 & I \end{bmatrix} - \begin{bmatrix} I - \Lambda & 0 \\ 0 & I - \Gamma \end{bmatrix} \begin{bmatrix} A & B \\ C & E \end{bmatrix}}_{J} \right)^{-1} \begin{bmatrix} \Lambda u^0 \\ \Gamma c^0 \end{bmatrix} \tag{17}$$

and from the series expansion of $(I-J)^{-1} \approx \sum_{k=0}^{\infty} J^k$, one gets that $u^* = \Lambda u(0) + (I-\Lambda)(Au(0) + Bc(0)) + ((I-\Lambda)A)^2 u(0) + (I-\Lambda)B(I-\Gamma)Cu(0) + (I-\Lambda)A(I-\Lambda)Bc(0) + (I-\Lambda)B(I-\Gamma)Ec(0) + h.o.t.$. We put the focus on the first order terms $(I-\Lambda)(Au(0)+Bc(0))$ and notice that $[A, B]\vec{1}_{N+M} = \vec{1}_N$, which in particular entails the equality constraint: $B_i + \sum_j A_{i,j} = 1$. Thus, by strengthening the influence of of the social network for user $i$, characterized by $\sum_j A_{i,j}$, the influence of the recommended content, characterized by $B_i$ needs to go lower. □

### F.2   PROOF OF LEMMA 1

*Proof.* We note that under a greedy RS for each user $i$, we have $r(u_i^t) = \min_j(||c_j^t - u_i^t||_2)$, and thus $u_i^t \in \mathcal{F}_j^t$. Furthermore, given $\Gamma = I_M$, we have $c_j^0 = c_j^t, \forall t$ and we simply write $c_j$.

**Induction Hypothesis** Assume for any user $i$, with opinion $u_i^t$, governed by the system dynamics in equation 1, that $u_i^t, u_i^{t-1} \in \mathcal{F}_j^t \times \mathcal{F}_j^{t-1}$ with $(u_i^t - c_j) = \alpha^t(u_i^{t-1} - c_j)$, $\alpha^t \in [\eta, 1]$, where $\eta = (1 - \Lambda_i)A_{ii} \in [0, 1]$. That is, $(u_i^t - c_j)$ and $(u_i^{t-1} - c_j)$ are parallel, point in the same direction, and $||u_i^t - c_j|| \le ||u_i^{t-1} - c_j||$.

**Induction Step** Using equation 1 we can write

$$u_i^t = (1 - \Lambda_i)(A_{ii}u_i^{t-1} + B_i c_j) + \Lambda_i u_i^0$$
$$u_i^{t+1} = (1 - \Lambda_i)(A_{ii}u_i^t + B_i c_j) + \Lambda_i u_i^0. \tag{18}$$

Where we used the fact that matrix $A$ is diagonal and that $u_i^t, u_i^{t-1} \in \mathcal{F}_j^t \times \mathcal{F}_j^{t-1}$. For notational simplicity, we omit the Kronecker product in equation 18. Using $\Lambda_i u_i^0 = u_i^t - (1 - \Lambda_i)(A_{ii}u_i^{t-1} + B_i c_j)$, we get

$$u_i^{t+1} = -(1 - \Lambda_i)A_{ii}(u_i^{t-1} - u_i^t) + u_i^t \Rightarrow$$
$$(u_i^{t+1} - c_j) = (u_i^t - c_j) - (1 - \Lambda_i)A_{ii}(u_i^{t-1} - u_i^t).$$

Intuitively, this states that the user opinions $u_i^{t-1}, u_i^t, u_i^{t+1}$, and the creators opinion $c_j$ lie on straight line. Now let $\eta = (1 - \Lambda_i)A_{ii} \in [0, 1]$, we can write

$$u_i^{t+1} - c_j = (u_i^t - c_j) - \eta(u_i^{t-1} - u_i^t)$$
$$= (u_i^t - c_j) - \eta((u_i^{t-1} - c_j) - (u_i^t - c_j))$$
$$= (1 + \eta)(u_i^t - c_j) - \eta(u_i^{t-1} - c_j)$$
$$= (1 + \eta)(u_i^t - c_j) - \eta/\alpha^t(u_i^t - c_j)$$
$$= (1 + \eta - \eta/\alpha^t)(u_i^t - c_j).$$

The fourth equality follows from the induction hypothesis, namely: $(u_i^t - c_j) = \alpha^t(u_i^{t-1} - c_j)$. Now we observe that

$$\alpha^t \in [\eta, 1] \Rightarrow (1 + \eta - \eta/\alpha^t) \in [\eta, 1].$$

Now let $\alpha^{t+1} = (1 + \eta - \eta/\alpha^t)$, which leads to the desired property: $u_i^{t+1} - c_j = \alpha^{t+1}(u_i^t - c_j)$, with $\alpha^{t+1} \in [\eta, 1]$, where $\eta = (1 - \Lambda_i)A_{ii}$. We can use this property to further deduce that $||u_i^{t+1} - c_j||_2 \leq ||u_i^t - c_j||_2$. Because all other creators are stubborn as well, this directly implies that under the greedy RS: $u_i^{t+1} \in F_j^{t+1}$.

**Base Case** Induction now follows by:

$$u_i^0 \in \mathcal{F}_j^0 \Rightarrow$$
$$u_i^1 = (1 - \Lambda_i)(A_{ii}u_i^0 + B_i c_j) + \Lambda_i u_i^0$$
$$u_i^1 = (1 - \Lambda_i)(A_{ii}u_i^0 + (1 - A_{ii})c_j) + \Lambda_i u_i^0$$
$$u_i^1 = ((1 - \Lambda_i)A_{ii} + \Lambda_i)u_i^0 + (1 - \Lambda_i)(1 - A_{ii})c_j \Rightarrow$$
$$u_i^1 - c_j = ((1 - \Lambda_i)A_{ii} + \Lambda_i)(u_i^0 - c_j).$$

The proof now follows by noting that: $\eta = (1 - \Lambda_i)A_{ii} \leq ((1 - \Lambda_i)A_{ii} + \Lambda_i) = \alpha^1 \leq 1$. We conclude $u_i^1 - c_j = \alpha^1(u_i^0 - c_j)$, with $\alpha^1 \in [\eta, 1]$, which given stubborn users, implies $u_i^0, u_i^1 \in \mathcal{F}_j^0 \times \mathcal{F}_j^1$. Thus the user partitions $\mathcal{F}_0, .., \mathcal{F}_M$ are in fact static under the greedy RS and for any user $i \in \mathcal{F}_j$, the distance to the creator $||u_i^t - c_j||_2$, decreases monotonically with $\alpha^t$. □

## F.3 PROOF OF LEMMA 2

For each user receiving the recommended content of a *d-hop socially-aware RS* as in Definition 8, the opinion dynamics of the user reads us

$$u_i^{t+1} = (1 - \lambda_i)\sum_{k=1}^N a_{ik}u_k^t + \lambda_i u_i^0 + \frac{b_i}{|\text{in}_i(d)|}\sum_{j \in \text{in}_i(d)} u_j^t.$$

One can notice how, in this case, the recommender system executes a driving force of each user's opinion towards the mean of opinion of their $d$-neighborhood. In particular, every time that the $d$-neighborhood includes a user $j$ such that $j \in \text{in}_i(d)$ and $j \notin F_\ell$, $i \in F_\ell$ then the *user silhouette* of user $i$, as per Definition 6 decreases as the minimum outer-cluster distance $b(u_i)$ cannot increase while the average intra-cluster distance cannot decrease. Therefore, by following a complementary reasoning as in Appendix F.2, the global clusterization from Definition 7 decreases.

## G RELATED WORK

The better position our paper, the following table provides a schematic summary of the related work, by classifying user and creators as Static (S) or Dynamic (D) and wheather they are seen as embedded in a Network (N) or seen as Isolated (I). For the Recommender System we distinguish if it is Fixed (F), namely taken from the literature, or Explicitly Designed (ED).

| | Users | Creators | Recommender System |
|---|---|---|---|
| Us | D,N | D, I | ED |
| Lin et al. (2024) | D,I | D,I | F |
| Rossi et al. (2022) | D,I | N/A | ED |
| Lanzetti et al. (2023) | D,I | N/A | ED |
| Dean et al. (2024b) | D,I | D,I | ED |
| Ma et al. (2008) | S,N | N/A | F |
| Chandrasekaran et al. (2024) | D,N | N/A | ED |
| Ziegler et al. (2005) | S,I | N/A | ED |
| Cheng et al. (2017) | S, I | N/A | ED |
| Zhang et al. (2023) | S,I | N/A | ED |
| Zhang & Hurley (2008) | S,I | N/A | ED |

