# OpenReview forum: "Socially-Aware Recommender Systems Mitigate Opinion Clusterization"
_ICLR.cc/2026/Conference — Submitted to ICLR 2026_

### Official Review · Reviewer_Si6k · 2025-10-25

**Soundness:** 3
**Presentation:** 2
**Contribution:** 3
**Rating:** 6
**Confidence:** 3

**Summary:**

This paper proposes a model to analyze the opinions of users and content creators in a social network, where their interactions are mediated by a recommender system.  Using this model, the authors theoretically analyze the effect of different recommendation strategies and propose a social network-aware recommender system that accounts for the interaction between users and creators to drive the social network towards a steady state opinion that is less clustered compared to e.g., the highly polarized status quo of existing social networks that primarily optimize recommendations for user engagement.

**Strengths:**

This work distinguishes itself from prior work by considering an interplay between users and content creators, where the content creators are actually strategic and adapt their material to grow their audience.

There are theoretical results to support the findings discussed in the experiments, which is a nice addition to the paper and helps to contextualize things.

The experimental results are strong and demonstrate the paper’s claim that a recommender system which optimizes for both user satisfaction and avoids clustering has a meaningful impact on both metrics, and captures a “middle ground” between myopically maximizing engagement and maximizing content diversity.

**Weaknesses:**

The definition of satisfaction (cumulative difference of opinion between users and creators they interact with over time) makes sense, but the connection between satisfaction (in this sense) and engagement is not completely clear to me — the paper cites a classic paper on confirmation bias to argue that high confirmation bias = high engagement, but I suspect that engagement on modern social media platforms may be more nuanced (e.g., strategically showing users some content they disagree with might spur engagement in the form of comments, arguments, etc.)

Writing is, at times, difficult to parse.  As one example, at the end of the introduction, the following sentence is particularly difficult for me to understand: “We argue that, being opinion polarization a collective phenomenon, in order for the RS to mitigate such undesired effect, enhancing content diversification is not enough without taking the social network into account.”

The network size considered (N = 600 users) is fairly small, and the initial topology is randomly generated.  Furthermore, the random generation process is not based on, and is not compared with, one of the many random graph models that are known to create "real-looking" networks (two examples of such models are the Newman-Watts-Strogatz [1] and Barabasi-Albert [2] models).
This is in contrast to other papers such as [3] that evaluate on real social network data sets, including a data set with thousands of users [4].

[1] Emergence of Scaling in Random Networks, Albert-Laszlo Barabasi and Reka Albert, 10.1126/science.286.5439.509
[2] Random Graph Models of Social Networks, Mark Newman and Duncan Watts and Steven Strogatz, 10.1073/pnas.012582999
[3] Local Edge Dynamics and Opinion Polarization, Nikita Bhalla and Adam Lechowicz and Cameron Musco, arXiv:2111.14020
[4] Stanford Large Network Dataset Collection (https://snap.stanford.edu/data/)

**Questions:**

It is convincing that society should care about opinion polarization and its effects, but it is not clear to me that the entities who control (the majority of) recommender systems care about it.  The work tries to address this by proposing a recommender system that maintains a high level of “user satisfaction,” arguing that this is a good proxy for optimizing for engagement.  In the experiments, the paper shows that a recommender system optimized for engagement increases clustering.

For a social media platform to adopt an alternative that reduces opinion clusterization, I would guess that one would have to demonstrate a recommender system with both high engagement and low clustering, which may be infeasible.  It is possible that public pressure or regulation could compel these platforms to act, but such reforms seem like they would be  challenging to verify/enforce.  Can you speculate about how this work would relate to efforts in the real world?

On line 55, I believe “harmuful” should be “harmful”.

The proposed socially-aware recommender system takes the network structure into account when making recommendations.  At the scale of large social networks, is scalability a concern for such an algorithm?  It may be slow/infeasible to exactly recover the mean opinion of the d-hop influencers of each user in a real-time recommender setting — some discussion on this point may be helpful.  Similarly, it may be useful to report runtime measurements for the standard (engagement maximizing) recommender system versus the proposed socially-aware system in the Appendix.

---

> ### Author Response · Authors · 2025-11-21
>
> **W1)** We fully agree that engagement on modern social media platforms is far more nuanced than in our framework. However we just wanted to stay as close as possible to the original work from Lin et al. “User-Creator Feature Polarization in Recommender Systems with Dual Influence,” NeurIPS 2024, where they treat the user as isolated and conclude that promoting diversity is not enough to counteract polarization.  In this regard, we wanted to emphasize the role of the social network in counteracting such undesired phenomena.
> However, the reviewer is absolutely right and in the revised manuscript, we will clarify that satisfaction in our model is not intended as a comprehensive measure of engagement, but rather as a tractable proxy capturing one well established mechanism about confirmation bias driven preferences. We will further emphasize that our theoretical conclusions concern this specific mechanism, and that additional engagement dynamics present in real platforms could lead to different behaviors.
>
> **W2)** We agree with the reviewer, that sentence would have benefit from more argumentation. The statement came from the inspirational work from Lin et al. “User-Creator Feature Polarization in Recommender Systems with Dual Influence,” NeurIPS 2024. In their model, users are treated as isolated, and the authors note that “due to dual influence, myopically optimizing recommendation diversity might hurt long-term creation diversity and result in polarization of the system.” Our intention was to stress the importance of the network in order to account for opinion polarization as a collective phenomenon for the population of users. We recognize the way the statement is posed is confusing and will provide further explaination in order to contextualize it.
>
> **W3)** Thanks for pointing this out. We  now incorporated the ego-Facebook dataset [1] into our paper, comprising a social network with 4039 users. We also extend our result to the opinion dimension $n=3$, visualize the cluster structure in 3D and through 2D projections, and evaluate performance using our proposed clusterization and engagement metrics.
> A figure displaying the resulting opinion landscape in 3-dimensions is given in Figure 1. in https://imgur.com/a/socially-aware-recommender-systems-additional-plots-HK2GEU8.
>
> **Q1)** This is a very interesting point, thanks for raising it. One thing one could say is that it is not entirely true that fairness aspects, in this case polarization mitigation, come to a compromise in terms of efficiency but rather that it is a matter of time scales. In the short term, you might be losing in term of profit (/\ engagement), but in the long run, promoting non-polarizing content might come with a radical change in how people engage with content and ultimately lead to an increse in profit  (/\ engagement) as well.
>
> **Q2)** Thank you. We will make the proposed correction.
>
> **Q3)** Thanks for this question. The time complexity scales with the size of the incoming region $d$ and the number of connections that users have, but not with the size of the network. This stems from the fact that each user recommendation is only a function of the opinions in the localized environment. Furthermore, the approach is parallelizable. That is, given the current opinion landscape, each user's recommendation can be computed independently of all other users' recommendations. We believe that the localized and parallel nature of our algorithm ensures favorable scalability properties.
>
> [1] Julian McAuley and Jure Leskovec. Learning to discover social circles in ego networks. NIPS’12, pp. 539–547, Red Hook, NY, USA, 2012. Curran Associates Inc

---

> > ### Comment · Reviewer_Si6k · 2025-11-25
> > **Reviewer Response**
> >
> > Dear authors,
> >
> > Thanks for your response and efforts, my questions have been addressed.  The addition of experiments on a much larger and real data set (ego-Facebook) is a nice addition to the paper.
> >
> > I maintain my positive score.

---

### Official Review · Reviewer_8hkx · 2025-10-27

**Soundness:** 4
**Presentation:** 3
**Contribution:** 3
**Rating:** 8
**Confidence:** 5

**Summary:**

The paper studies a three-sided system involving users, content creators, and a recommender system (RS). At each timestep, both user and creator opinions co-evolve. Users are influenced by two factors: their social network (connected users), and the content they consume from creators. Content creators, in turn, are influenced primarily by user opinions.

The recommender system matches creators to users using a top-k closest-opinion criterion.
The authors analyze how different recommender strategies shape the system’s long-term dynamics: A greedy recommender, which optimizes purely for user satisfaction, tends to drive cluster formation (polarization) among creators. A non-greedy recommender can reduce this clusterization but at the cost of lower user satisfaction. Finally, the authors propose a socially aware recommender, denoted RS(d), which introduces a tunable parameter d — the number of hops in the social network considered for matching. This allows the system to interpolate between maximizing user satisfaction (d=0) and minimizing clusterization (larger d).

**Strengths:**

The paper makes an original contribution to the study of opinion dynamics in systems involving users, content creators, and recommender algorithms. While prior work such as Lin et al. has modeled the co-evolution of user and creator opinions, the introduction of user–user interactions through a social network makes the modelling more realistic. The model formulation is clear and well-motivated.

The socially aware recommender system, RS(d), that can interpolate between maximizing user satisfaction and reducing clusterization by tuning the parameter d (the number of hops in the social network) is both elegant and conceptually sound.
The main technical contributions, especially in Sections 4.1.2 and 4.2, are well executed. The authors show that:

- A greedy recommender focused solely on user satisfaction leads to creator clusterization;
- A non-greedy recommender reduces clusterization but lowers user satisfaction; and
- The proposed RS(d) strikes a balance between these extremes, offering a controllable trade-off.

**Weaknesses:**

The modeling and results are insightful, but there are areas where the presentation and clarity should be improved:

1.  The main contribution statements.
The description of the paper’s key findings in the introduction and contribution section could be made more precise and technical. For example:
- The statement “We demonstrate that opinion clusterization is positively correlated with the influence of the RS” could be revised to:
 “We show that a recommender system (RS) that greedily optimizes for user satisfaction leads to opinion cluster formation among creators.”
- Similarly, instead of “We provide a new optimization-based RS that explicitly incorporates social connections to reduce clusterization effects while keeping a high level of user satisfaction,” a clearer version might be:
 “We propose a social-network–aware recommender, RS(d), where the parameter d (number of user hops) controls the trade-off between user satisfaction and the extent of creator clustering, with low d leading to higher satisfaction but more clusters, and high d reducing clusters at the cost of satisfaction.”



2) Improve the organization of related work.
 The related work section would benefit from a comparison table (either in the main text or appendix). The table could clearly indicate for each prior work whether users and creators are modeled as static or dynamic, and whether the recommender system is fixed or explicitly designed.
This addition would make it easier to identify the specific novelty of this paper which is the introduction of user–user social interactions and the co-evolution of user opinions influenced by both their social network and content creators.
For example, Lin et al. include a helpful summary table in their appendix that clarifies which axes of the opinion dynamics problem each prior work explored. Adopting a similar approach here would highlight the distinct contributions of this work and improve readability.

**Questions:**

Q1. What happens to the evolution of creator opinions in the special case where
$(I_M​−T)E=0$, i.e., when creators evolve solely based on user opinion feedback without creator coupling? Do the qualitative results presented in Section 4 (particularly regarding clusterization and satisfaction trade-offs) still hold under this condition, or does the system converge differently?

Q2. In Equation (13), the user opinion is defined as the average opinion of all users within d hops, and this aggregated opinion u_i is then used for top-k creator selection. In the limiting case where d is very large, all users would share approximately the same aggregated opinion and hence receive similar recommendations. How does this affect the evolution of creator opinions?

---

> ### Author Response · Authors · 2025-11-21
>
> Thanks for the good assessment of our paper.
>
> **W1)** thanks for your suggestions that enhance the clarity of our contributions. We will be more than happy to implement them in the final version of our manuscript.
>
> **W2)** Thank you for the insightful suggestion. We agree that adding a table would significantly improve the clarity of our comparison with the existing literature by highlighting both similarities and differences. We will be happy to add it to the appendix in the final version of our manuscript.
>
> **Q1)** We point out that both $E$ and $(I_M-T)$ are taken as diagonal matrix and therefore in the present setting there is already no coupling among creators. The diagonal elements of the aforementioned matrices just serve as a scaling factors on how fast creators'opinions evolve.
>
> **Q2)** Thank you for this insightful question. In a fully connected graph, the reference recommendation converges to the same value for all users in the limit (d very large). When the opinion landscape is balanced (i.e., the average user opinion is close to the origin), all users receive recommendations of creators closest to the origin.
> For users we expect this to lead to a stable configuration where opinions disperse around the origin over time, with the degree of concentration depending on stubbornness. Less stubborn users converge closer to the neutral position.
>     For creators this mechanism produces a "winner-takes-it-all" dynamic. creators positioned near the origin receive the majority of interactions as they consistently appear in top-$k$ recommendations, while those farther from the origin receive minimal engagement and consequently do not adapt their opinions.
>     In summary: choosing a very large $d$ causes the Recommender System to suggest creators who are neutral relative to the global opinion landscape. Since all users then receive similar recommendations, the system cannot sustain polarization as they converge toward homogeneous opinions by interacting with the same set of creators. Meanwhile, creators positioned at the neutral point dominate engagement, while creators with polar opinions remain marginalized with negligible influence on the system dynamics.
>     Thanks for the very insightful question.

---

### Official Review · Reviewer_jobP · 2025-10-31

**Soundness:** 2
**Presentation:** 2
**Contribution:** 3
**Rating:** 4
**Confidence:** 4

**Summary:**

The user-creator feedback interaction is known to cause filter bubbles and polarization in recommender systems.  This paper develops a social-network-aware recommender system that explicitly accounts for the user-creator feedback interaction and exploits the topology of the social network to promote diversity of the system.  In particular, the paper proposes to recommend to a user contents that are close to the average of the user's neighbors. Simulations show that this method reduces polarization/clusterization, while maintaining a certain level of user satisfaction.

**Strengths:**

(S1) This paper introduces a new perspective to the study of user-creator feedback interaction in recommender systems: social network.  In addition to the recommended content, users' opinions are also affected by their neighbors on the social network.  This paper shows that such social network structure can be leveraged to reduce polarization.  This is an interesting observation.  It is a good positive contribution to the largely negative literature on polarization in recommender systems.

**Weaknesses:**

However, I have major concerns about the theoretical rigor of this work.

(W1) **Lack of a formal definition of “influence” in Theorem 1**.  Theorem 1 claims that "increasing the influence of A reduces the influence of B". Here, A and B are matrices describing social and recommender effects, but the paper never defines what "influence" means in a quantitative sense.  Although the notation suggests that A and B affect users’ steady-state opinions, the authors do not provide a scalar metric (e.g., a norm, spectral quantity, or sensitivity measure) that maps these matrices to real-valued influences. As a result, the statement “increase of A decreases B” is ambiguous and cannot be rigorously evaluated. The theorem would benefit from a precise mathematical formulation of “influence” and clear assumptions under which this complementarity holds.

(W2) **Key proofs didn't consider the steady-state of dynamics**.  The proofs of Lemma 1, Corollary 1, and Lemma 2 analyze user-creator interactions without explicitly accounting for the steady-state of the dynamic system.  Since both users and creators evolve under feedback loops and social influence (the Friedkin-Johnsen model), the asymptotic behavior (i.e., equilibrium or stability conditions) is critical to determining long-term clusterization outcomes. However, the proofs in Appendix E appear to rely on instantaneous or static relationships rather than steady-state analysis of the full dynamics.  Without establishing convergence properties or equilibrium characterizations, these results may not generalize to the long-run behavior claimed in the paper.



Another concern:

(W3) The proposed "socially-aware recommender" recommends contents that are close to the average of the neighbors of a user.  How is this method compared to practices like "your friends may like these" recommendations, and other methods in the literature that consider users' social network?  Such comparisons are missing.

**Questions:**

## Questions for the authors

(Q1)  See (W3).



## Suggestions

* Typo: line 055: "harmuful" -> "harmful"
* Typo: Line 189: what is $\hat A$ ?
* Some references are outdated or inaccurate:
  * [Eilat & Rosenfeld, arXiv 2023] (Performative recommendation: diversifying content via strategic incentives) should be [Eilat & Rosenfeld, ICML 2023]
  * [Hron et al, arXiv 2022] (Modeling Content Creator Incentives on Algorithm-Curated Platforms) should be [Hron et al, ICLR 2023]
  * [Lin et al, NeurIPS 2025] (User-Creator Feature Polarization in Recommender Systems with Dual Influence) should be [Lin et al, NeurIPS 2024]

---

> ### Author Response · Authors · 2025-11-21
>
> **W1)** We agree that the theorem should be framed more rigorously. The word influence is used in two related but distinct senses in this work and must be clarified. First, inter-agent influence denotes how users affect one another through the social network and is modeled via the Friedkin–Johnsen dynamics (the matrix $A$). Second, recommender influence denotes the direct effect that the recommendation algorithm has on a user's opinion (the matrix
> $B$). In the theorem the expression influence refers to the overall impact of the social network dynamics on a user's steady-state opinion. Informally, the result states that, when more attention (weight) is assigned to other users' contents (an increase in inter-agent influence), the relative effect that the recommender system can exert on the user's long-run opinion decreases. This relationship is formalized under the assumption that the concatenated matrix $[A,B]$ is row-stochastic, which captures a finite attention budget that each user allocates between peers and recommendations (see e.g. [1] for the attention-budget interpretation). We will make the theorem statement more rigorous in this regard and make sure this aspect gets clarified in text in the final version of our manuscript.
>
>
>
> **W2)** Thanks for this very insightful question.
>
> Corollary 1 provides a condition that holds at each time step, independent of any asymptotic or steady-state considerations. Given our engagement metric, any recommendation that differs from the creator closest to the user (according to the L2 norm of the difference in their opinions) represents a suboptimal choice from a recommender perspective that tries to maximize for engagement, both at steady state and point-wise.
>
> Theorem 1 guarantees convergence to a steady state in the opinion landscape, we forgot to mention it in the proof of the teorem and gave it as a granted that limits exists, the proof for the steady state behaviour follows the same steps as Theorem 2 in [2]. Thanks for pointing this out. We will make sure to mention it in the final version of the paper. A refined version of Lemma 1 and 2 that explicitly characterizes the monotonic temporal increase in clusterization in Lemma 1 and monotonic temporal decrease in Lemma 2—can be combined with Theorem 1 to extend these results to steady-state behavior. Thanks for pointing this out, we will make sure to incorporate these insights.
>
> **W3)** We thank the reviewer for raising this important point. Although our socially-aware recommender uses information from a user’s social network, it differs substantially from standard “your friends may like these’’ mechanisms and from prior social-aware RS (which are referred as "Social RS" in our introduction) in both purpose and design.
> Classical social RS typically use social links as an additional signal for preference prediction. In that literature, social information is incorporated into collaborative filtering or matrix-factorization models to improve accuracy or personalization. The objective is to recommend items that a user is more likely to engage with by exploiting (static) correlations between friends’ ratings or interactions.
> In contrast, our method does not use neighbors’ opinions as a predictor of future preferences. Instead, we dinamically exploit the structure of the social network as a control tool to shape the long-term opinion dynamics in a way that mitigates opinion clusterization.
> Therefore, while  our method also uses social information, its motivation, formulation, and effects differ fundamentally from “your friends may like these’’ approaches.
> We will revise the related-work section to clarify this distinction and include a dedicated comparison in the final version.
>
> **S1)** Thanks, will correct this.
>
> **S2)** That's a typo. Will replace with $A$. Thank you.
>
> **S3)** Thanks for suggesting the updated version. We will replace them in the final version.
>
>
> [1]G. Cocca, P. Frasca, C. Ravazzi,"A Coupled Friedkin–Johnsen Model of
> Popularity Dynamics in Social Media", LCSS 2025
>
> [2] B. Sprenger, G. De Pasquale, R. Soloperto, J. Lygeros, F. Dörfler, "Control Strategies for recommendation systems in social networks", LCSS 2024

---

> > ### Comment · Reviewer_jobP · 2025-11-25
> >
> > **Further question on (W1):**
> > Let me clarify my question.  I understand that $A$ and $B$ are matrices capturing the inter-agent effect and creator-user effect.  Your Theorem 1 says "increasing the influence of $A$ (on the static state) decreases the influence of B (on the static state)".  In Appendix E.1, you said that the static state is $(I - \Lambda) (A u(0) + B c(0))$.  Now my question is, when you say "increasing the influence of $A$", do you mean increasing each entry of $A$, or increasing the norm of the vector $A u(0)$, or something else?  Note that $A$ is a matrix, and the $A u(0)$ is a vector.  It is unclear to me what you meant by increasing the influence of some matrix on some vector.
> >
> > **Further question (W2):**
> > I know that you want to say that properties (such as clusterization increase) that hold at every time step should imply the same properties for the static states.  However, that is not immediate. Consider two dynamics d1(t) and d2(t).  Initially, d1(0) is the same as d2(0).  After one step, d1(1) is better than d2(1), e.g., smaller clusterization.  But then how do you argue that d1(2) is still better than d2(2) ?  If d1(t) = d2(t) at time step t, then by your per-step argument, we have d1(t+1) better than d2(t+1) at step t+1.  However, now d1(t+1) is different from d2(t+1), so you cannot apply the per-step argument to go from t+1 to t+2 --- it is not guaranteed that d1(t+2) is still better than d2(t+2).  In other words, you cannot easily compare two dynamics starting from different initial states.  It is unclear to me how your per-step argument implies any properties about the limit static state.
> >
> > **Apply to all comments:**
> > Note that ICLR allows uploading a revised version of the PDF.  If you think the problem can be addressed formally, could you please update the PDF?

---

> > > ### Author Response · Authors · 2025-12-02
> > >
> > > We thank the reviewer for the two important follow-up questions.
> > >
> > > **W1)** We have now included a remark in Theorem 1 explaining what "influence" refers to (see updated version). In summary, for each user $i$, the FJ model enforces a row stochastic constraint: $\sum_j A_{ij} + B_i = 1$. The influence of the social connections is captured by $\sum_j A_{ij}$, while the influence from the recommender system is captured by $B_i$. Thus, both the social network and the recommender system are complementary, as increasing the summed weight of the social network (i.e., $\sum_j A_{ij}$) inevitably decreases the weight from the recommender system ($B_i$). This model captures the idea of "limited attention span." Intuitively, the more one engages with their peers the less it does with online personalized recommendations.
> > >
> > > **W2)** We agree with the reviewer's observation that the consequence of "a piecewise increment in clusterization" towards the steady state is not immediate. We have thus rewritten Lemma 1 with a stronger focus on the monotone decrease of user and creator distance (according to the L2 norm). The resulting lemma now bears a direct relation to Theorem 1, which captures the steady-state behavior of the user dynamics. In the new proof, the (time-varying) convergence rate can be stated explicitly.
> > > For the non-greedy (i.e., a socially aware) recommender system, a steady state cannot be guaranteed in general. We provide the following insight:
> > >
> > > Let $R$ be a recommender system with an adapting strategy. That is, $R$ can choose at each instant if it wants to be greedy ($d=0$) or socially aware ($d>0$). We know that choosing $d=0$ will increase clusterization while $d>0$ will decrease it. The opposite holds for satisfaction: $d=0$ will maximize satisfaction while $d>0$ will be suboptimal. This is the pointwise property in our discussion. Now we know that if $R$ chooses to keep the strategy greedy ($d=0$), then the users will converge to the closest creator and reach a clusterized steady state. On the other hand, if $R$ chooses to keep the strategy social ($d>0$), then we cannot guarantee a steady state, but we know that at any time step, the socially aware choice ($d>0$) will lead to a less clusterized environment at the next time step as opposed to a greedy one.
> > > If we interpret correctly, your final question boils down to: What happens if $R$ decides to follow a socially aware strategy for $T$ time steps, after which it decides to initiate a greedy strategy? Given Lemma 1 and Theorem 1, we know that a clusterized steady state will be reached. But how is this steady state different from the steady state that would have been reached if $R$ decided to follow a greedy strategy from the beginning?
> > > Due to the complex nature of the network environment under a socially aware recommender (time-varying graph that induces time-varying system dynamics), it turns out to be very hard to provide generalized statements about the overall system behavior and it definitely leaves space for thoughts on how our proposed algorithm can be improved to have formal guarantees on the steady state as well.
> > >
> > > We thank the reviewer for their very insightful questions that helped us improve the quality of our work. For ease of review, the added text appears in blue in the newly uploaded PDF.

---

### Official Review · Reviewer_ydWZ · 2025-10-31

**Soundness:** 2
**Presentation:** 3
**Contribution:** 3
**Rating:** 4
**Confidence:** 2

**Summary:**

This paper introduces a closed-loop model of users, creators, and recommender systems using multi-topic opinion dynamics. It theoretically shows that greedy recommendation increases opinion clusterization, and proposes a socially-aware d-hop recommender that leverages neighborhood opinions to balance satisfaction and polarization.

**Strengths:**

1. Clear formulation of a closed-loop social–RS dynamic.
2. Provides a simple, interpretable design knob (d-hop neighborhood).
3. Theoretical complementarity between social and RS influence is insightful.
4. Experiments illustrate a meaningful satisfaction–clusterization trade-off.

**Weaknesses:**

1. Theoretical results rely on static user partitions, deterministic recommendations (k=1), and diagonal social influence matrices—settings that largely remove true social interactions. Lemma-level assumptions are not empirically validated, and the gap between deterministic theory and stochastic simulations (softmax sampling) remains unaddressed.
2. All experiments use synthetic networks constructed from initial opinion similarity, which risks circular reasoning regarding clusterization. Only 2-D opinion space is tested, with no validation on real or semi-synthetic social graphs or content interaction data.
3. The paper does not compare with standard diversification or exposure-aware recommenders (e.g., MMR, xQuAD, calibrated ranking). It is unclear whether the proposed d-hop social averaging offers benefits beyond existing diversity mechanisms.
4. Clusterization is measured solely via the silhouette coefficient on k-means clusters. Alternative graph-level or diversity metrics (e.g., assortativity, modularity, diversity@k) are not reported, and sensitivity to the clustering hyperparameters is not discussed.
5. Results are shown only for a few d and k values; key parameters and noise robustness are not systematically explored.

**Questions:**

see Weakness.

---

> ### Author Response · Authors · 2025-11-21
>
> **W1)** Thanks for pointing this out. We believe that the assumption made for the theoretical analysis do not comprimise the essence of the conclusions we draw. Assumptions about staticity, stationarity, deterministicity have been made in order to simplify the setting that otherwhise would have required to deal with time varying, stochastic matrices which would have required dealing with a lot of mathematical details that would have distracted the focus from what is the essence of the phenomena our model aims to capture.
>
> **W2)**  To strengthen the message of our paper we have now incorporated the Ego-Facebook dataset [1], comprising a real social network with 4039 users. We present the resulting opinion landscape for opinion dimension $n=3$ and visualize the cluster structure in 3D and through 2D projections, see Figure 1.: https://imgur.com/a/socially-aware-recommender-systems-additional-plots-HK2GEU8. We evaluate performance using our proposed clusterization and engagement metrics.
>
> **W3)** Our core finding, that incorporating social awareness into recommender systems can help mitigate polarization, is compatible with a broad range of diversity-aware recommenders. In particular, any diversification/recommender mechanism that operates on feature representations can be extended to include social awareness by computing the average feature representation of users within a d-hop neighborhood. We adopt top-k recommendation as our base mechanism because it is well-established [2],[3], widely used in practice, and provides a clear framework for illustrating our results. Our goal is therefore not to position our approach as outperforming existing  diversity promoting recommendation mechanisms, but rather to introduce a strategy that can be integrated with them.
>
> **W4)** Social networks exhibit homophily [4]. That is, people tend to connect with people with similar opinions. Thus,  graph-level diversity metrics are not well suited for mesuring clusterization in our setting where the considered social network is hompophilic, and hence by design it  exhibits strong homophilic properties and thus also high modularity and assortativity.
>
> We chose the silhouette coefficient on k-means clusters as it offers a well established and easy interpretable way to measure clusterization from a macro perspective, regardless the actual structure of the social network.
>
>
> **W5)** Variation of d is discussed for values in between 0 and 6. as can be seen in figure 5. and 6. in https://imgur.com/a/socially-aware-recommender-systems-additional-plots-HK2GEU8. One can notice that both clusterization and engagement saturate after d=3 or d=4 depending on the average user's in-degree. Thus the authors believe that the the chosen interval encompasses all important results and no additional insights can be generated by varying the localized region more.
>
> For the synthetic dataset, we discuss the variation of parameter $k$ (recommendations per user) for k in {1,2,3,4,5}. Given the pool of 50 creators, this covers up to 10 percent of creators, thus the authors believe this is a reasonable range for evaluation and increasing going beyond 10\% of the creators population would be quite unrealistic. Figure 2-4 at https://imgur.com/a/socially-aware-recommender-systems-additional-plots-HK2GEU8, display the influence of choosing larger or smaller values for k. choosing larger K reduces the number of clusters but is not sufficient to counter clusterization effectively.
>
> Furthermore we provide discussions regarding the variation of interaction parameters in the user-creator and creator-user dynamics. Figure 5. and 6. in https://imgur.com/a/socially-aware-recommender-systems-additional-plots-HK2GEU8, show the difference in engagement and clusterization between a social network with little to medium connections between users (figure 5.) and social network with many connections between users (Figure 6.). Clearly, increasing connections between users reduce clusterization effects. The fact that strong social ties reduce clusterization is in fact mimicked by our RS strategy and shown by our theoretical results (Theorem 1).
>
> [1] Julian McAuley and Jure Leskovec. Learning to discover social circles in ego networks. Neural Information Processing Systems - Volume 1, NIPS’12
>
> [2] Paul Covington, Jay Adams, and Emre Sargin. Deep Neural Networks for YouTube Recommendations. 10th ACM Conference on Recommender Systems
>
> [3] Tao Lin, Kun Jin, Andrew Estornell, Xiaoying Zhang, Yiling Chen, and Yang Liu. User-creator
> feature polarization in recommender systems with dual influence. NeurIPS
>
> [4] Miller McPherson, Lynn Smith-Lovin, and James M. Cook. Birds of a feather: Homophily in social
> networks. Annual Review of Sociology, 27:415–444, 2001

---

### Official Review · Reviewer_F6js · 2025-10-31

**Soundness:** 2
**Presentation:** 2
**Contribution:** 2
**Rating:** 4
**Confidence:** 3

**Summary:**

This paper studies how recommender systems that optimize for engagement can increase opinion polarization and clusterization. The authors introduce a socially-aware recommender system that explicitly incorporates users’ social network structure into the recommendation process. They model the joint dynamics of users, content creators, and the RS using an extended FJ opinion dynamics model. It presents some theoretical results showing that social influence and recommendation influence can act as contrary forces.

**Strengths:**

- The idea of theoretically analyzing the interaction between recommendation and social networks under FJ model (with high-dimensional opinions) is novel.
- The paper is theoretically solid, despite some simplified assumptions.
- The studied topic is still of great importance nowadays.

**Weaknesses:**

1. I think this paper has not phrased its main contributions accurately. It claims to have developed a socially-aware recommendation system, but this idea is not novel. Most mainstream social media platform these days such as Meta and LinkedIn have used social networks in their recommendations with algorithms taking care of topic diversification and polarization. The proposed recommendation system in Section is more of a high-level idea sketch rather than any real system that can operate on real-world data. I think the main novelty of this paper, instead, is that it analyzes the relationship between engagement-based recommendation with polarization, under a theoretical framework, which yields some theoretical insights for industrial practice. It is very important in this regard to clarify the scope and limitation of any conclusions made in this paper.
2. The experiment is very limited. The data is small, synthesized in a naive way, and only one experiment is presented. Please significantly expand the experiment section.
3. Discussion of many related works that study the relationship between recommendation and polarization under FJ framework is missing, for example, [1-3].
4. There are many typos in the paper: “explotied” → “exploited”, “harmuful” → “harmful”, “explicitely” → “explicitly”, “recieve” → “receive”, “deigns” → “designs”, “sweetspot” → “sweet spot”.

[1] On the Relationship Between Relevance and Conflict in Online Social Link Recommendations, NeurIPS 2023.

[2] Minimizing Polarization and Disagreement in Social Networks via Link Recommendation, NeurIPS 2021.

[3] Towards consensus: Reducing polarization by perturbing social networks, IEEE Transactions on Network Science and Engineering.

**Questions:**

Please address the Weaknesses above.

---

> ### Author Response · Authors · 2025-11-21
>
> Thank you for the kind words of appreciation.
>
> **W1)** We agree with the reviewer’s observation that real-world social media platforms already incorporate information from a user’s ego-network when generating recommendations. Our original emphasis on social-network awareness as a way to address opinion polarization was motivated by the framework of Lin et al., “User-Creator Feature Polarization in Recommender Systems with Dual Influence,” NeurIPS 2024. In their model, users are treated as isolated, and the authors note that “due to dual influence, myopically optimizing recommendation diversity might hurt long-term creation diversity and result in polarization of the system.” Our intention was to highlight that, in contrast, incorporating social-network structure can serve as a natural mechanism for mitigating such polarization.
>     That said, we acknowledge the reviewer’s point that the main focus of our contribution may not have been sufficiently clear. In the revision, we will explicitly emphasize that our primary contribution is theoretical: we characterize the relationship between content personalization and opinion polarization. We will also clarify that our model is a stylized abstraction. While we believe it captures the core dynamics of real-world recommender systems, actual platforms may exhibit additional behaviors due to human irrationality, heterogeneous engagement incentives, or mechanisms beyond confirmation bias.
>
> **W2)** We agree with the reviewer's observation that relying solely on one experiment with synthetic data limits the scope of the work. We   have now incorporated the ego-Facebook dataset [1] into our paper, comprising a social network with 4039 users. Beside considering a real dataset we also extend over opinion dimensions to $n=3$ topics and visualize the cluster structure in 3D and through 2D projections as displayed in Fig 1. https://imgur.com/a/socially-aware-recommender-systems-additional-plots-HK2GEU8
>
> **W3)** Thanks for pointing these references out. We will be sure to mention them and add a dedicate paragraph in the Related Work section on the role of link recommendations in mitigating polarization.
>
> **W4)** Thank you for pointing these typos out. We will correct them and carefully revise the entire paper.
> \end{itemize}
>
> [1] Julian McAuley and Jure Leskovec. Learning to discover social circles in ego networks. Neural Information Processing Systems - Volume 1, NIPS’12

---

### Meta-Review · Area_Chair_qeT7 · 2026-01-07

**Summary:**

After carefully checking the paper, the reviews, the rebuttal, and the author-reviewer discussions, I think the weak points outweight the strong points. Two reviewers acknowledge the authors' work, but the other reviewers raised concerns regarding the paper’s novelty, theoretical rigor, and experimental issues. After carefully reviewing the manuscript, I believe the authors still have not addressed the limitations related to dynamic scenarios, nor the insufficiency of the experiments. The weaknesses are not likely to be fixed in the camera-ready version. Thus, I recommend rejecting this paper.

**Reviewer Concerns:**

The score remains unchanged. I believe the authors have addressed the issues related to experimental hyperparameters, but have not resolved the concerns regarding theoretical rigor and more generalizable experimental validation. I have carefully read the rebuttal. The rebuttal does not address any important concern raised by the reviewers.

**Reviewer Scores:**

The score remains unchanged. I have carefully read the rebuttal. The rebuttal does not address any important concern raised by the reviewers.

---

### Decision · Program_Chairs · 2026-01-26

Reject